# Intramolecular OHO Hydrogen Bonding in Dibenzoylmethane Enol: Raman Spectroscopic and Quantum Chemical Study

**Boris A. Kolesov** [1,*]**, Elena A. Pritchina** [2] **and Aleksey Ya. Tikhonov** [3]

[1]   A.V. Nikolaev Institute of Inorganic Chemistry, Siberian Branch, Russian Academy of Sciences, 630090 Novosibirsk, Russia

[2]   Faculty of Natural Sciences, Novosibirsk National Research State University, 630090 Novosibirsk, Russia

[3]   N.N. Vorozhtsov Novosibirsk Institute of Organic Chemistry, Siberian Branch, Russian Academy of Sciences, 630090 Novosibirsk, Russia

[*]   Correspondence: kolesov@niic.nsc.ru

**Abstract:** In the present work, the intramolecular O-H···O hydrogen bonding in 3-hydroxy-1,3-diphenylprop-2-en-1-one (keto-enol form of dibenzoylmethane, DBM) was investigated. For this purpose, the Raman spectra of polycrystalline samples of ordinary (H-DBM) and deuterated (D-DBM) 3-hydroxy-1,3-diphenylprop-2-en-1-one in the temperature range of 5–300 K were measured. It was found that low-temperature hydrogen bonding is extremely strong, the proton and deuteron are located in the midpoint of the O···O segment, and their ground and first excited vibrational states are located above the barrier $U_0$ between the local minima. The vibrational frequencies, in this case, are 1543 and 1709 cm$^{-1}$ for the proton and 1045 and 1087 cm$^{-1}$ for the deuteron. As the temperature rises and the barrier height increases in H-DBM, the zero-point vibrational state of the proton begins to move into one of the local minima at T > 50 K, while the excited state remains in the broad single-well potential. The same is observed in D-DBM, but with a significant temperature delay. Compounds with donor ($-OCH_3$) and acceptor ($-NO_2$) substituents in the phenyl ring were also synthesized and their spectra were obtained. Both results confirm existing ideas about the nature of the extremely strong hydrogen bond. The quantum-chemical calculation of the vibrational spectrum of H-DBM and D-DBM is consistent with the experimental results.

**Keywords:** β-diketones; intramolecular hydrogen bonding; infrared spectroscopy; Raman spectroscopy; isotope effects; quantum-chemical calculation





## 1. Introduction

The spectral manifestation and diagnosis of strong and extremely strong O−H···O hydrogen bonds is one of the interesting and intriguing problems of vibrational spectroscopy.

A necessary condition for the occurrence of a strong hydrogen bond is the high electronegativity of both atoms, donor X and acceptor Y [1]. In this case, the proton experiences a strong interaction with both the donor and the acceptor, resulting in a short X···Y distance. When a proton is displaced from a donor to an acceptor, its potential energy in any case is described by a double-well potential. If the X and Y atoms are identical, which most likely occurs for the O−H···O bond, then the proton potential becomes symmetrical double-well, and the height of the barrier between the potential minima near the donor and acceptor atoms depends on the distance O···O and for a strong bond becomes comparable with the energy of the vibrational states of the proton, zeroth, and first excited. When the barrier height is less than the energy of the zeroth vibrational state, the potential becomes single-well, and the H-bond becomes extremely strong. For a strong bond, the zeroth vibrational state of the proton remains in a local minimum near one of the oxygen atoms, and the first excited state passes into the combined potential, in which the proton should be in the middle of the O···O segment. Since the transition from zeroth to the first excited vibrational state must be accompanied by a change in the proton coordinate, Raman

scattering is forbidden for strong H-bonds. In the case of an extremely strong H-bond, both vibrational states of the proton are in the combined potential (the proton is in the middle of the segment O···O), and the vibrational transition in the Raman spectra becomes active. In other words, only states with extremely strong hydrogen bonds can be observed in Raman spectra, and there is no way to estimate the relative number of O−H···O and O···H···O states.

The conditions of high electronegativity of the donor and acceptor, as well as their identity, are easily realized for intramolecular H-bonds, in which the bond is a part of symmetrical organic cycles.

For example, the widely known and studied dibenzoylmethane molecule, DBM (Figure 1), may well form a strong or extremely strong intramolecular H-bond, since the enol cycle can become symmetric under certain conditions and the electronic shells of both oxygen atoms can require saturation. Nevertheless, numerous studies of the structural [2–5] and vibrational [6–8] properties of DBM-enol have found no direct evidence for the presence of strong or extremely strong H-bonds. It should be noted, however, that the experiments in all the cited works were performed at room temperature. It was shown in [1] that the specificity of strong and extremely strong H-bonds is that their spectral detection often becomes possible only at low temperatures. The purpose of this work is to study the strength of the hydrogen O-H···O bond in DBM. For this reason, this work presents a Raman spectral study of polycrystalline samples of H-DBM and D-DBM in the temperature region of 5–300 K. For both types of compounds, normal and deuterated, quantum-chemical calculations of the energy states and vibrational spectrum were performed. The spectra of compounds with donor and acceptor substituents in the phenyl ring are also presented.

**Figure 1.** Graphical scheme of studied compounds DBM (R = H), DBM-OMe (R = OCH$_3$), and DBM-NO$_2$ (R = NO$_2$). (**A**,**B**) are two symmetrical positions of the proton on an intramolecular hydrogen bond.

## 2. Experimental

### 2.1. Preparation and Deuterization

The methods of the synthesis of β-diketones are represented in Refs. [9,10].

We used Aldrich's H-DBM, 98%, Vinyl acetate (99%) from Steinheim (Germany) and Chemical Line's Oxalyl chloride (99%) from St. Petersburg (Russia) without additional purification. The remaining reagents and solvents used in the work were of Russian production.

Preparation of D-DBM. To a solution of H-DBM (0.224 g, 1.0 mmol) in dry dioxane (3 mL), D$_2$O was added (0.663 g, 33 mmol, deuterium content 96.5%) and heated at 62–65 °C for 2 h without adding NaOD [11]. The solution was evaporated in a rotary evaporator to dryness. Dry dioxane (3 mL) and D$_2$O (0.663 g, 33 mmol) were added to the obtained crystalline precipitate and heated at 62–65 °C for 2 h and again evaporated to dryness. This procedure was repeated four more times. The resulting crystalline precipitate was kept in a vacuum until reaching a constant weight. According to the $^1$H NMR spectrum, there is a substitution of hydrogen atoms in the DBM-enol form (reduction in the signal intensity of

the C-H-enol form at 6.85 m.p. in CDCl$_3$). The residual hydrogen content in the obtained D-DBM was not more than 10%.

3-Hydroxy-1,3-bis (4-methoxyphenyl) prop-2-en-1-one (keto-enol form of 1,3-bis (4-methoxybenzoyl) methane, DBM-OMe, m. p. 117–118 °C [12]) and 3-hydroxy-1,3-bis (4-nitrophenyl) prop-2-en-1-one (keto-enol form of 1,3-bis (4-nitrobenzoyl) methane, DBM-NO$_2$, m. p. 247–249 °C [13]) were synthesized according to Refs. [12,13], respectively.

## 2.2. *Quantum Chemical Calculations*

The geometries and harmonic vibrational frequencies of dibenzoylmethane in enol form and the transition state for the movement of protons between two oxygens were calculated in a gas phase at the M06-2X [14]/6-311+(d,p) [15], B3LYP/6-311+(d,p) [16], and wB97XD/aug-cc-pVTZ [17,18] levels of theory. For all calculations, the GAUSSIAN09 suite of programs [19] was employed (Gaussian 09, Revision A.01; Fox, Gaussian, Inc.: Wallingford, CT, USA, 2009). Since all three levels of the theory give close values of vibrational frequencies, in what follows, we will use only M06-2X/6-311+(d,p).

## 2.3. *Raman Spectroscopy*

The Raman spectra were collected on a LabRAM Horiba single-stage spectrometer with a CCD Symphony (Jobin Yvon) detector with 2048 horizontal pixels. The laser power (633 nm line of He-Ne laser) of the sample was typically less than 0.1 mW. The spectra at all temperatures were measured in backscattering collection geometry with a Raman microscope. The powder was fixed on the cold finger of the cryostat. The spectral resolution was 0.7 cm$^{-1}$.

## 3. Results

Keto-enol forms of DBM crystallize in several different polymorphic modifications [4,5]. The samples are single-phase and their phase composition is identical. The diffraction patterns of H-DBM and D-DBM coincide and fully correspond to the crystal structures of a stable orthorhombic polymorph (I), Pbca, Z-8, a = 10.853 (1), b = 24.441 (1), c = 8.7559 (1) Å [20–22].

The IR spectra of H-DFP and D-DFP at room temperature are widely known in the literature (see, for example, Refs. [6,7]), do not contain pronounced separate bands of hydrogen bond vibrations, and are not presented here. It can only be noted that the package of vibrational bands in the 1400−1700 cm$^{-1}$ region in H-DBM admits the presence of a broad band overlapping with all others, and the presence of a broad weak band at ~1100 cm$^{-1}$ in D-DBM is possible.

Figure 2a shows the Raman spectra of H-DBM at various temperatures. The spectra contain two striking features, namely, the bands at 1543 and 1709 cm$^{-1}$, which are intense at low temperatures but practically disappear at room temperature. In deuterated D-DBM crystals (Figure 2b) both of these modes are not observed at all, but two new groups of bands appear: in the region of 1000–1100 and 1400–1550 cm$^{-1}$. The dependence of their intensities on temperature significantly differs from that observed in H-DBM: they do not decrease to zero with increasing temperature, but only slightly begin to decrease at T > ~200 K.

Table 1 lists the scheme and calculated frequency of only those vibrations of the DBM enol fragment that are related to hydrogen bonding in H-DBM and D-DBM for the case of a strong hydrogen bond, in which the proton and deuteron are in the middle of the segment connecting the donor and acceptor oxygen atoms. The remaining modes of the vibrational spectrum of DBM belong mainly to the well-known vibrations of phenyl rings in the region of 600–1600 cm$^{-1}$, as well as torsional and bending vibrations in the region of 100–600 cm$^{-1}$. In addition, Table 1 also shows the vibrational frequencies of the enol ring in the case of the H-bond, in which the hydrogen atom is located in a minimum near one of the oxygen atoms. However, it should be kept in mind that, since the symmetry of the molecule changes when the proton is displaced from the center of the H-bond to the oxygen atom, there is no direct correspondence between the vibrations in these two cases,

and many modes characteristic of the symmetric cycle are absent in the asymmetric cycle. Moreover, the calculated frequencies of the proton vibrations when both states, ground and exited, are localized in the minimum near the donor oxygen are values of ~2400 cm$^{-1}$. Bands with such frequencies are observed neither in the IR nor in the Raman spectra of the compounds.

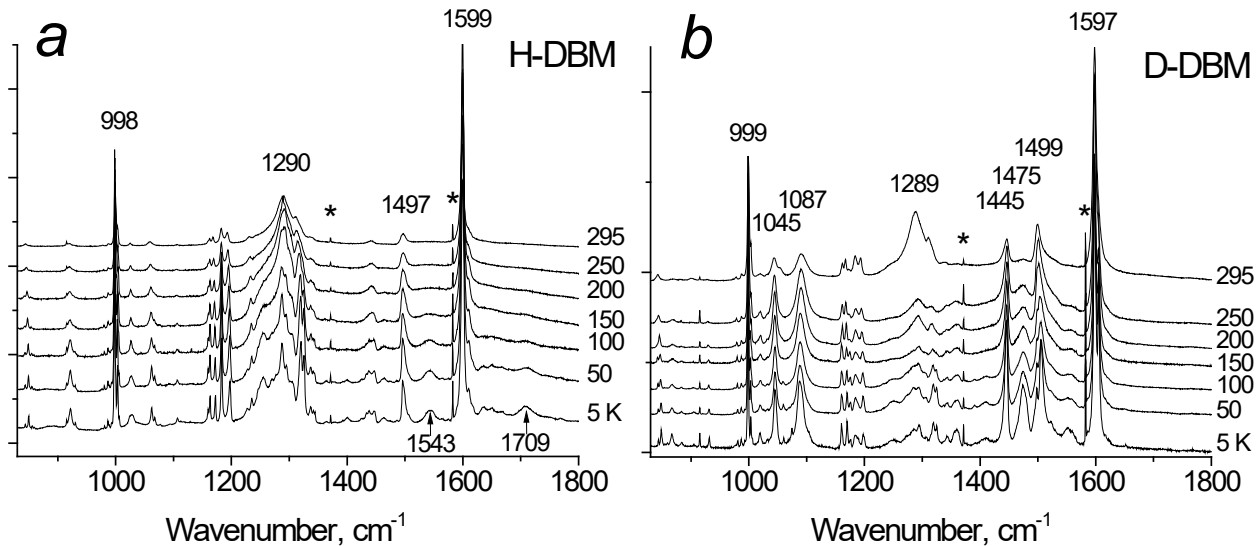

**Figure 2.** (**a**,**b**) The spectra of H−DBM (**a**) and D−DBM (**b**) at different temperatures in the range of the hydrogen bond vibrations. The asterisk (*) marks the positions of the line of the Ne lamp, which was registered at each temperature for the wavenumber correction of the spectra.

**Table 1.** Calculated with M06-2X/6-311+G(d,p), the frequency of vibrations in 3-hydroxy-1,3-diphenylprop-2-en-1-one, including the displacement of the proton of the enol ring.

| | | H-DBM | | | D-DBM | | |
|---|---|---|---|---|---|---|---|
| | | Exper. | Calc. | | Exper. | Calc. | |
| | | | O⋯H⋯O | O−H⋯O | | O⋯D⋯O | O−D⋯O |
| 1 | Ph, Ph, O−H−O structure | 682 | 652 | 935 | 675 | 651 | |
| 2 | Ph, Ph, O−H−O structure | 791 | 776 | 696 | 786 | 775 | 427 486 |
| 3 | Ph, Ph, O−H−O structure | 998 | 979 | | 998 | 951 | |

**Table 1.** *Cont.*

| | | H-DBM | | | D-DBM | | |
|---|---|---|---|---|---|---|---|
| | | Exper. | Calc. | | Exper. | Calc. | |
| | | | O⋯H⋯O | O−H⋯O | | O⋯D⋯O | O−D⋯O |
| 4 | Ph, Ph, O, O, H±, Out of plane | ? | 1274 | 960 | ? | 931 | 674 |
| 5 | Ph, Ph, O, O, H | 1543 | 1602 | 1382 | 1045 | 1477 | |
| 6 | Ph, Ph, O, O, H | 1599 | 1615 | | 1613 | 1615 | |
| 7 | Ph, Ph, O, H, O, + vibr. of Ph | 1613 | 1667 | | ? | 1541 | 1426 |
| 8 | P, P, O, H, O, + vibr. of Ph | 1640 | 1671 | | ? | 1671 | 1645 1697 |
| 9 | Ph, Ph, O, O, H | 1709 | 1897 | 1638 1664 | 1087 | 1294 | 883 1077 1114 |

## 4. Discussion

Since the O⋯O distance in H-DBM is 2.46 Å at room temperature [3,5,20–22], this H-bond belongs to the strong ones. It was shown in [1] that in strong O-H⋯O hydrogen bonds, the zeroth vibrational state of the proton remains in the local minimum near the donor atom, while the first excited state goes into a broad potential well that combines both minima and forms a single-well potential. In this case, scattering by proton vibrations is prohibited, since the transition to an excited vibrational state requires a change in the proton coordinates on the bond.

However, as the temperature decreases, the vibrational states of the molecule (mainly low-frequency ones) freeze out, and the H-bond shortens. This leads to a hardening in the H-bond and a decrease in the barrier between local minima at the donor and acceptor atoms. As a result, both vibrational states of the proton, the ground state and the first excited state, find themselves in a broad unified potential, which corresponds to the case of an extremely strong hydrogen bond, and the vibrational activity of the proton in the Raman spectrum is restored. This is precisely the process observed in Figure 2a, where the two modes, 1543 and 1709 cm$^{-1}$, change in intensity from zero at room temperature

(scattering is forbidden) to a finite value at 5 K. In the calculated spectrum, these modes correspond to vibrations 5 and 9 (see Table 1). The proton vibration frequencies indicate the formation of an extremely strong hydrogen bond, which is stronger than has been known in the literature so far (see, for example, Ref. [1] for the intermolecular H-bond in dimethylformamide). Thus, the proton in the H-DBM is at the middle point of the hydrogen bond O···H···O at a low temperature (5–50 K) and begins to gradually localize near one of the oxygen atoms when the temperature rises from 50 K.

In the D-DBM samples, the 1045 and 1087 $cm^{-1}$ modes are assigned to the deuteron vibrations (Figure 2b). Their intensity also changes with temperature but is much weaker than in H-DBM. Thus, a slight decrease in the intensity of these modes begins only at T > 200 K and remains high at room temperature. Consequently, the potential barrier $U_0$ between local minima in D-DBM is much lower than in H-DBM with respect to the energy of vibrational states of the deuteron, and both of them are in a wide single-well potential in almost the entire temperature range. It is only at T > 200 K that the value $U_0$ of the potential barrier between the minima is compared with the energy of the ground vibrational state.

The difference in the position of the vibrational states of the proton and deuteron with respect to the barrier height $U_0$ between the local minima lies in the value of $U_0$ itself. The last value for a symmetric double-well potential is defined as [1]

$$U_0 = k\Delta^2, \tag{1}$$

where $k$ is the force constant of the O-H bond, and $\Delta$ is the distance between local minima in oxygen atoms. The quantum uncertainty of the coordinate of the particle of mass $m$ is proportional to $m^{-1/4}$, and for the deuteron, it is less than for the proton [23]. It, at the same distance between the oxygen atoms for intramolecular H-bonds, leads to a smaller value of hydrogen bonding, i.e., the smaller force constant $k$ for deuteron. A consequence of this circumstance is both the smaller value of $U_0$ in the D-DBM compared to $U_0$ in the H-DBM, and the location of the vibrational states of the proton and deuteron relative to the barrier $U_0$. In addition, the smaller force constant in D-DBM than in H-DBM means that the shift in the vibrational frequency of the proton when it is replaced by deuterium will be larger than in $\sqrt{2}$. Indeed, the experimental frequency shift of mode 5 is 1543(H)/1045(D) = 1.48 and that of mode 9 is 1712(H)/1088(D) = 1.57. However, the anomalous shift of the vibrational frequency during deuteration can be observed only in compounds with strong and extremely strong H-bonding.

To further substantiate the above concepts, compounds with donor (-$OCH_3$) and acceptor (-$NO_2$) substituents in the phenyl ring were synthesized and their spectra were obtained. It was assumed that the additional electron density from the donor substituent migrates to the e-cycle, reaches the oxygen atoms of the H-bond, and reduces their electronegativity. This should yield a weakening of the H-bond. Conversely, the depletion of the electron density in the oxygen atoms of the e-cycle in the case of substituents of the acceptor type (-$NO_2$) will serve to increase their electronegativity and strengthen the H-bond. Figure 3a shows the DBM-OMe spectra at various temperatures. At low temperatures, a broad intense band at ~1000 $cm^{-1}$ is formed in the spectra, which is related to vibrations of the H-bond. Figure 3b shows similar spectra of DBM-$NO_2$. They do not have strong spectral indications of hydrogen bond vibrations, although the weak mode at 1563 $cm^{-1}$ at T = 5 K may be related to this vibration.

Now it is necessary to find out how the H-bond strength changes upon the introduction of donor and acceptor substituents. It is known that the vibration frequency of the terminal C=O group involved in the formation of the H-bond is a sensitive indicator of the strength of the latter.

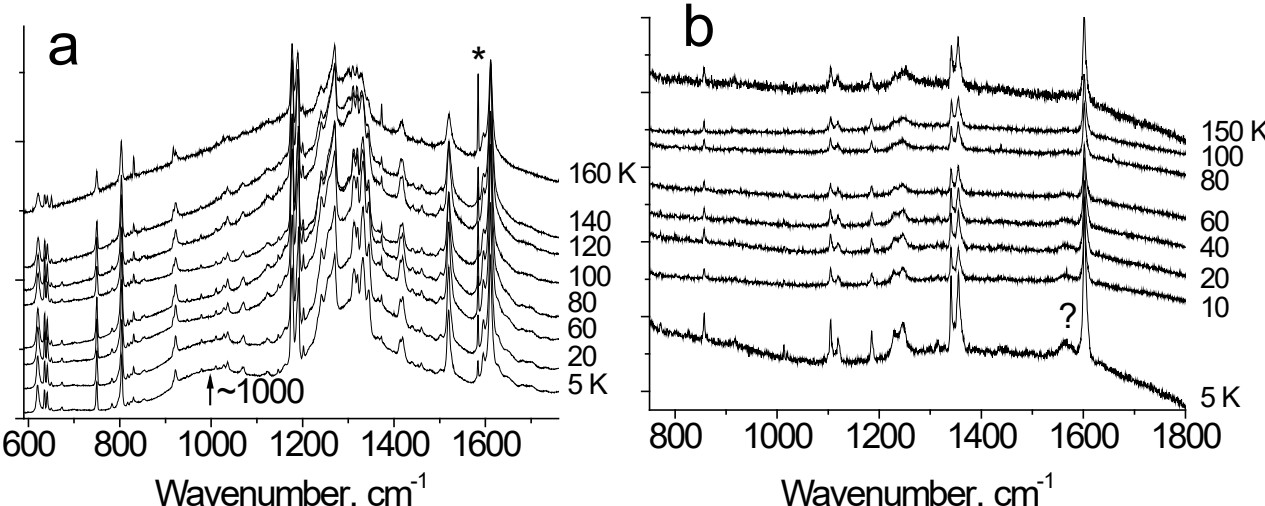

**Figure 3.** (**a**,**b**) Raman spectra of DBM−OMe (**a**) and DBM−NO₂ (**b**) at different temperatures. The spectra of both compounds contain a broad luminescent background. The asterisk (*) marks the positions of the line of the Ne lamp, which was registered at each temperature for the wavenumber correction of the spectra. The sign (?) refers to a weak line, the origin of which is unclear.

Figure 4 shows the Raman spectra at room temperature of all three compounds in the range of vibrational frequencies of the C=O bonds of the e-cycle. The ~1600 cm$^{-1}$ mode is shifted to the high-frequency in DBM-OMe up to 1608 cm$^{-1}$ compared to 1598 cm$^{-1}$ in DBM and is practically not shifted in DBM-NO₂. This means that the H-bond in DBM-OMe is weaker and does not change in DBM-NO₂ compared to that of DBM. In other words, a weaker H-bond corresponds to a lower frequency of O-H vibrations. This result is an indisputable confirmation of the dependence of the H-bond vibration frequency in strong and extremely strong hydrogen O-H⋯O bonds, according to which the O-H vibration frequency increases with the increasing strength of the H-bond (Figure 9 in Ref. [1]). The absence of a visible result at the spectral level in DBM-NO₂ compared to the unsubstituted compound may indicate that the strength of the H-bond in DBM has reached its limit.

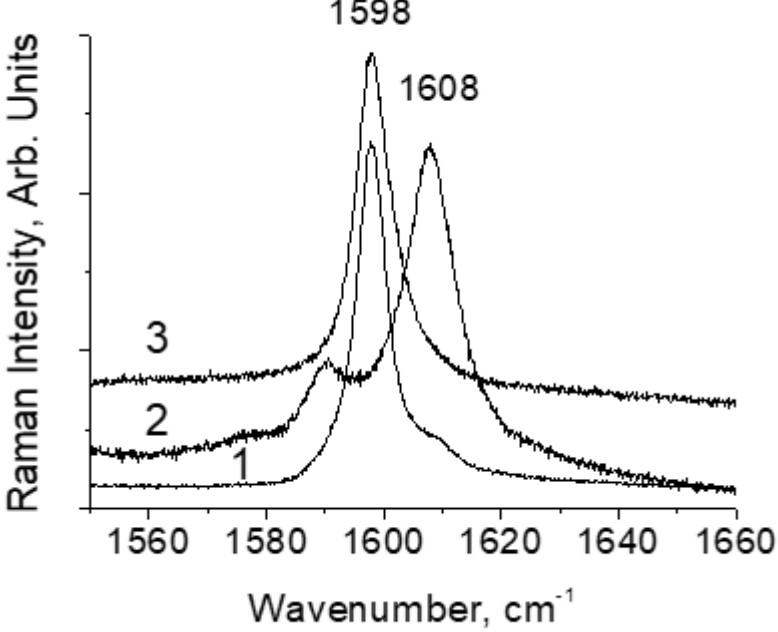

**Figure 4.** Room temperature Raman spectra of DBM (**1**), DBM-OMe (**2**), and DBM-NO₂ (**3**) compounds in the range of vibrational frequencies of the C=O bonds of the e-cycle.

The location of the proton (deuteron) in the middle of the O···O segment means the delocalization of molecular orbitals in the enol ring, just as it happens in ordinary aromatic compounds. Conversely, the localization of the proton (deuteron) in one of the minima near the oxygen atom means the appearance of alternating single and double CO bonds in the enol cycle. Since in the latter case, the transition of a proton from one local minimum to a neighboring one does not change the energy of the e-cycle (if we abstract from the possible interaction of protons of neighboring DBM molecules in the crystal lattice), we should observe proton jumps in H-DBM as the temperature increases. As was shown in Ref. [24], due to proton jumps and the proton-induced changes in the multiplicity (and length!) of $C-O$, $C=O$ bonds of the e-cycle from single to double give rise to the same anharmonic processes that are observed in solids when the temperature increases, i.e., a decrease in the bond vibration frequency. Figure 5a shows the dependence of the $C=O$ bonds of the e-cycle vibrational frequency at 1599 cm$^{-1}$ mode in H-DBM on temperature. It can be seen that the active process of proton hopping on the $O-H\cdots O$ bond begins at T~100 K. A similar vibrational mode of 1445 cm$^{-1}$ in the D-DBM shows, on the contrary, a slight increase in frequency with increasing temperature (Figure 5b). Of course, this effect is due to the weakening of the O···D···O hydrogen bond with increasing temperature and the absence of deuteron jumps on bonds (or their very low rate).

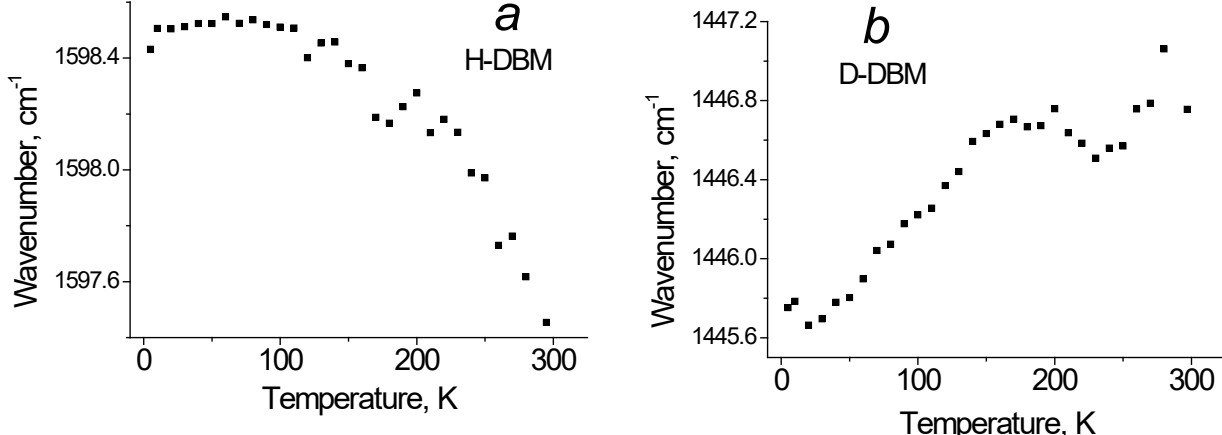

**Figure 5.** (**a**,**b**) Temperature dependence of the position of 1599 cm$^{-1}$ in H-DBM (**a**) and the 1445 cm$^{-1}$ D-DBM (**b**) band peak of the C=O stretching vibration of the enol cycle.

Figure 6 shows the spectra of various compounds with extremely strong H-bonds. In this series, the weakest H-bond is observed in Me-DBM and the strongest in DBM. Dimethylformamide (DMF)$_2$H, whose spectrum was taken from Ref. [25], occupies an intermediate position. The general trend is as follows: with an increase in the rigidity of the hydrogen bond, the frequency of proton vibrations increases, while the integrated intensity and bandwidth decrease. The behavior of the frequency of proton vibrations on the extremely strong H-bond, i.e., for O···O distances less than 2.5 Å, has already been discussed above. The band width is possibly related to the position of the energy of zero-point proton vibrations $\omega_0$ to the height of the barrier between two symmetrical potential minima characterizing the extremely strong hydrogen bond [1]. The closer $\omega_0$ to the barrier height, the more uncertain its value and the wider the vibration band in the spectrum. As for the integral intensity, this issue remains unclear and requires further research.

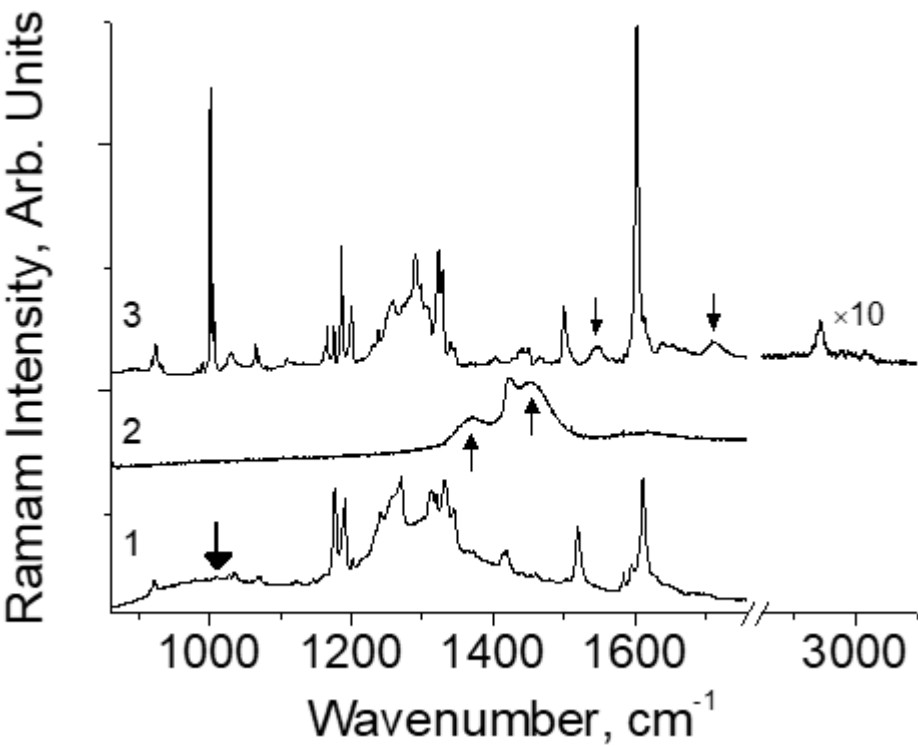

**Figure 6.** Raman spectra of Me-DBM (**1**), (DMF)$_2$H (**2**), and DBM (**3**) at 5 K in the range of hydrogen bond vibrations. The positions of the H-bond bands are marked with arrows.

In this work, the calculation of the vibrational spectrum of molecules was carried out within the framework of the standard approach (see section "Quantum chemical calculations"). However, it was shown in Ref. [26] that the experimental frequencies of proton vibrations on the hydrogen bond turn out to be substantially lower than those given by calculations. The reason for the discrepancy between the experiment and the calculation lies in the anharmonicity of hydrogen bond vibrations. Therefore, to assess the applicability of quantum chemical calculations for systems with strong hydrogen bonds, it is necessary to consider in general terms the nature of hydrogen bond anharmonicity.

The hydrogen bond is fundamentally anharmonic, i.e., the potential energy of a proton on a hydrogen bond along the donor-acceptor segment fundamentally and significantly differs from the quadratic dependence. The evolution of the proton potential energy curve as the H-bond changes from weak to extremely strong is shown in Ref. [1] (Figures 1–4 in [1]). If for a weak H-bond the anharmonicity value, i.e., the deviation of the proton potential energy curve from the parabola is small (Figure 1 in [1]), then for the moderate bond the anharmonicity becomes maximum (Figure 2 in [1]). In other words, the greater the anharmonicity of a hydrogen bond, the stronger the bond and, accordingly, the deviation of the proton vibrational frequency from its maximum value at ~3600 cm$^{-1}$. However, in this case, it is not the frequency of proton vibrations that determines the anharmonicity of the bond, but, on the contrary, the magnitude of the anharmonicity determines the observed frequency of proton vibrations. The calculation of the frequencies of proton vibrations for weak and moderate H-bonds according to the standard procedure, which does not fully take into account anharmonic effects, is fraught with error the greater and stronger the H-bond.

The situation, however, changes dramatically upon passing from moderate H-bonds to strong and extremely strong ones ($d_{O\cdots O} < 2.5$ Å, Figure 4 in Ref. [1]). In this case, the right and left branches of the potential curve are symmetrical with respect to each other, and in this respect, the potential is similar to the harmonic potential. However, when the zeroth vibrations energy of the proton becomes higher than the barrier height, the bottom of the proton potential well is almost flat, since it is formed by two closely spaced potential

minima, one from the donor and the other from the acceptor. As a result, the proton potential curve for an extremely strong H-bond becomes similar to the "particle in in the box" potential. The anharmonicity of the "particle in in the box" potential is also large [27], but its sign is the opposite of the anharmonicity sign for the common single-well potential (i.e., 6–12 potential of Lennard-Jones). In other words, if the energies of stationary states in a single-well potential, as a rule, approach each other with an increase in the vibrational quantum number *n*, then they diverge in a "particle in in the box" potential. At present, it is impossible to say for certain what value of anharmonicity is realized for an extremely strong hydrogen bond. It is clear, however, that the dependence of the anharmonicity on the rigidity of the hydrogen bond is not monotonic and requires a separate study.

It should also be noted that strong and extremely strong H-bonds are quite rare in chemistry and are poorly understood, and this article is just aimed at filling the gap in our knowledge about the properties of strong and extremely strong OHO bonds.

## 5. Conclusions

The specificity of the intramolecular hydrogen bonding is that the donor–acceptor distance cannot be any, as in intermolecular H-bonds. It is determined by the geometric parameters of the molecule or molecular cycle in which the H-bond is realized. In this case, the donor–acceptor distance may be short enough to promote the formation of an extremely strong H-bond. This work establishes that the H-bond in DBM refers to an extremely strong O-H···O hydrogen bond. However, the often double-well shape of the proton potential along the strong H-bond and the associated need to change the localization of the proton upon vibration excitation make it difficult to detect intramolecular H-bonds in vibrational spectra at room temperature. For this reason, to reliably record intramolecular H-bonds, it is necessary to resort to low-temperature measurements.

**Author Contributions:** Conceptualization, investigation, writing, B.A.K., investigation, E.A.P., investigation, A.Y.T. All authors have read and agreed to the published version of the manuscript.

**Funding:** The research was supported by the Ministry of Science and Higher Education of the Russian Federation, N 121031700314-5 and 1021051403061-8-1.4.1 (NIOC SB RAS).

**Institutional Review Board Statement:** Not applicable.

**Informed Consent Statement:** Not applicable.

**Data Availability Statement:** No data supporting reported results.

**Acknowledgments:** The authors would like also to acknowledge the Multi-Access Chemical Research Center SB RAS for spectral and analytical measurements.

**Conflicts of Interest:** The authors declare no conflict of interest.

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
