# Peer review of "Intramolecular OHO Hydrogen Bonding in Dibenzoylmethane Enol: Raman Spectroscopic and Quantum Chemical Study"

_2673-8023, doi:10.3390/micro3010009_

Round 1

Reviewer 1 Report

The manuscript by Kolesov and co-workers reports a Raman-spectroscopy study of the symmetry of an intramolecular hydrogen bond in 3-hydroxy-1,3-di-phenylprop-2-en-1-one. Using the selection rules of Raman spectroscopy, one can experimentally find the conditions under which this hydrogen bond becomes symmetrical and single-well in the ground vibrational state of this molecule. In particular, the study describes the effect of temperature and H/D isotopic substitution on the symmetry of this hydrogen bond. The topic of the study is relevant in the field and its motivation is justified.

However, the manuscript has a number of shortcomings.

1.     “A necessary condition for the occurrence of a strong hydrogen bond is the high electronegativity of both atoms, donor X and acceptor Y [1].” This requirement is neither sufficient nor the most important. The donor X and acceptor Y must have the same electronegativity. In this case, high electronegativity will even limit the strength of the hydrogen bond (see e.g. doi.org/10.1021/jp5082033 and doi.org/10.1016/j.molstruc.2011.12.027).

2.     “The proton energy along the straight line connecting the donor and acceptor is described in any case by a double-well potential.” Please correct the sentence. The proton coordinate must not necessarily follow “the straight line connecting the donor and acceptor” whereas a single-well asymmetric potential is possible when the electronegativities of X and Y are different.

3.     The authors should discuss other systems with strong intramolecular hydrogen bonds. One of the closest examples is hydrogen maleate anion (see e.g. doi.org/10.1021/jacs.9b08492, doi.org/10.1021/ja00202a050, doi.org/10.1021/ja962662n, doi.org/10.1021/ja0017615).

Author Response

Reviewer 1.

Thank you very much for your valuable comments.

Comment:

“A necessary condition for the occurrence of a strong hydrogen bond is the high electronegativity of both atoms, donor X and acceptor Y [1].” This requirement is neither sufficient nor the most important. The donor X and acceptor Y must have the same electronegativity. In this case, high electronegativity will even limit the strength of the hydrogen bond (see e.g. doi.org/10.1021/jp5082033 and doi.org/10.1016/j.molstruc.2011.12.027).

Response:

In the works indicated by the reviewer, the N-H×××N hydrogen bond was studied for compounds dissolved in aprotic polar solvents, in which the solvent molecules of course interact with the proton of the H-bond and distort it. As for the influence of the electronegativity of donor and acceptor atoms on the rigidity of the H-bond, the best example in this matter is liquid or crystalline water, in which the electronegativity of the oxygen atoms of the donor and acceptor is fundamentally the same and is very small due to the saturation of their electron shell. As a result, the O-H×××O hydrogen bond in water is one of the weakest in nature.

Comment:

“The proton energy along the straight line connecting the donor and acceptor is described in any case by a double-well potential.” Please correct the sentence. The proton coordinate must not necessarily follow “the straight line connecting the donor and acceptor” whereas a single-well asymmetric potential is possible when the electronegativities of X and Y are different.

Response:

Done

Comment:

The authors should discuss other systems with strong intramolecular hydrogen bonds. One of the closest examples is hydrogen maleate anion (see e.g. doi.org/10.1021/jacs.9b08492, doi.org/10.1021/ja00202a050, doi.org/10.1021/ja962662n, doi.org/10.1021/ja0017615).

Response:

Our next work will focus on these compounds.

Reviewer 2 Report

The presented manuscript can be published after some corrections.

The manuscript should be completed with the information what kind of tautomeric form is observed under the experimental conditions.

It is worth to add a scheme containing the studied compounds and, consequently, the abbreviations used in the manuscript.

It is necessary to add the region of 1800 - 4000 cm-1 into the paper.

The experimental section needs to be completed with NMR spectra of the synthesized compounds as a proof of purity of these compounds.

In the paper the authors stress not once that the studied hydrogen bond is the strongest. In this connection they should also cite the following from the literature: the length of the hydrogen bridge and the data of neutron diffraction for the proton position.

The data of Isotopic Spectroscopic Ratio should be given for all the studied compounds.

The manuscript hardly contains the calculations analysis. I recommend to remove the part of the paper concerning the calculations (pp.3 and 10, Quantum chemical calculations).

The list of the literature should be edited according to the requirements of the journal.

Author Response

Reviewer 2.

Thank you very much for your valuable comments.

Comment:

        The manuscript should be completed with the information what kind of tautomeric form is observed under the experimental conditions.

Response:

The molecules of DBM and its derivatives are symmetrical with respect to the permutation of the proton on the hydrogen bond, so the left and right tautomers are indistinguishable and must be present in an equiprobable amount. At low temperatures, when the proton is in the middle of the hydrogen bond, tautomerism disappears.

Comment:

It is worth to add a scheme containing the studied compounds and, consequently, the abbreviations used in the manuscript.

Response:

Done

Comment:

It is necessary to add the region of 1800 - 4000 cm-1 into the paper.

Response:

Done (see Fig. 6 in the revised text).

Comment:

The experimental section needs to be completed with NMR spectra of the synthesized compounds as a proof of purity of these compounds.

Response:

All compounds used in the work were prepared on the basis of purchased components with guaranteed purity (see section “Preparation and deuterization”).

Comment:

In the paper the authors stress not once that the studied hydrogen bond is the strongest. In this connection they should also cite the following from the literature: the length of the hydrogen bridge and the data of neutron diffraction for the proton position.

Response:

The hydrogen bond length in DBM is given in the “Discussion” section (first line). The reference refers to 5 papers, one of which, [22], is based on neutron diffraction data.

Comment:

The data of Isotopic Spectroscopic Ratio should be given for all the studied compounds.

Response:

The authors fully agree with the comment of the reviewer. However, isotope substitution of DBM-OMe and DBM-NO2 will not add critical information, and we did not do this to optimize the time needed to complete the article.

Comment:

The manuscript hardly contains the calculations analysis. I recommend to remove the part of the paper concerning the calculations (pp.3 and 10, Quantum chemical calculations).

Response:

We do not agree with the comment of the reviewer. The calculation of the vibrational spectrum contains important information necessary for assigning vibrational lines in the spectrum, and this information is used in the text.

Comment:

The list of the literature should be edited according to the requirements of the journal.

Response:

Done